# Hypoxic Conditions Modulate Chondrogenesis through the Circadian Clock: The Role of Hypoxia-Inducible Factor-1α

**DOI:** 10.3390/cells13060512

**Published:** 2024-03-14

**Authors:** Krisztián Zoltán Juhász, Tibor Hajdú, Patrik Kovács, Judit Vágó, Csaba Matta, Roland Takács

**Affiliations:** Department of Anatomy, Histology and Embryology, Faculty of Medicine, University of Debrecen, H-4032 Debrecen, Hungary; juhasz.krisztian@anat.med.unideb.hu (K.Z.J.); hajdu.tibor@med.unideb.hu (T.H.); patrik.kovacs@med.unideb.hu (P.K.); vago.judit@med.unideb.hu (J.V.)

**Keywords:** chondrogenesis, hypoxia, circadian clock, transcription factor, osteoarthritis, HIF-1

## Abstract

Hypoxia-inducible factor-1 (HIF-1) is a heterodimer transcription factor composed of an alpha and a beta subunit. HIF-1α is a master regulator of cellular response to hypoxia by activating the transcription of genes that facilitate metabolic adaptation to hypoxia. Since chondrocytes in mature articular cartilage reside in a hypoxic environment, HIF-1α plays an important role in chondrogenesis and in the physiological lifecycle of articular cartilage. Accumulating evidence suggests interactions between the HIF pathways and the circadian clock. The circadian clock is an emerging regulator in both developing and mature chondrocytes. However, how circadian rhythm is established during the early steps of cartilage formation and through what signaling pathways it promotes the healthy chondrocyte phenotype is still not entirely known. This narrative review aims to deliver a concise analysis of the existing understanding of the dynamic interplay between HIF-1α and the molecular clock in chondrocytes, in states of both health and disease, while also incorporating creative interpretations. We explore diverse hypotheses regarding the intricate interactions among these pathways and propose relevant therapeutic strategies for cartilage disorders such as osteoarthritis.

## 1. Introduction

Hyaline cartilage is the most abundant type of cartilage in the human body and is present in areas such as the trachea and the bronchi, nose, epiphyseal growth plate, sternum, ventral segments of the ribs, and synovial joints. The primary function of articular cartilage is to provide a resilient surface with minimal friction in almost every synovial joint in the body, where it plays a crucial role in resisting compressive and shear forces [1].

Maintaining the organized architecture of articular cartilage is essential for its function. Damage to articular cartilage can lead to musculoskeletal conditions. Osteoarthritis (OA) is the most common form of chronic inflammatory joint diseases (arthritis), and a leading cause of musculoskeletal disability worldwide [2]. OA is a whole joint disease, involving all joint tissues, such as the articular cartilage, synovial membrane, subchondral bone, meniscus, and infrapatellar fat pad [3], and is characterized by the progressive degeneration of articular cartilage. However, articular cartilage has a limited capacity for regeneration [2]. As articular cartilage degenerates, symptoms such as joint pain, swelling, stiffness, and loss of joint movement arise. OA can affect any joint but most commonly impacts the knee, the hip, the spine, and the joints of the hand [4]. A combination of factors, such as joint structure and function, the weight-bearing nature of articular cartilage, and the specific type of mechanical load, contribute to why OA predominantly affects specific joints while sparing others. Ankle OA, for example, is often secondary to factors such as trauma, chronic ankle instability, malalignment, and arthropathies [5]. OA is a heterogeneous disease with multiple etiologies, clinical phenotypes, and molecular endotypes, which necessitates differential targeting approaches, opening pathways for the development of effective disease-modifying OA drugs (DMOADs) [6].

Despite the socio-economic burden posed by OA, typical management is currently palliative and reactive, rather than proactive and preventive [2]. Joint replacement surgery is a clinically relevant procedure for end-stage OA, but it is associated with more serious adverse events compared to non-surgical treatment options [7]. However, there are no currently available surgery-based, material-based, cell-based, or drug-based treatment options that could reliably restore the structure and function of hyaline articular cartilage, despite extensive research and recent developments. There is an urgent need for fundamental research in this field to better understand why the regeneration of articular cartilage fails [8].

Cartilage tissue engineering can be effectively utilized in both managing and preventing OA by employing various strategies. These include using cell populations capable of regenerating the cartilage matrix, designing suitable scaffolds to support cell growth and phenotype, incorporating growth factors to promote cartilage regeneration, and applying mechanical stimuli to enhance the effectiveness of the engineered cartilage [9]. Cartilage tissue engineering has promising prospects, especially when chondroprogenitor cells (CPC) are implanted in the site of injury [8]. This has resulted in the effective filling of the defect in animal [10] and in human pilot studies [11]. However, a better understanding of the chondrogenic pathways is necessary to assist the delivered cells in forming and sustaining neocartilage at a site where the inflammatory microenvironment is not optimal for supporting chondrogenic differentiation.

Chondrogenic pathways are regulated by multiple external and internal factors [12]. Biological rhythms are known to influence cartilage biology, and disruptions to this rhythmicity have been reported to be risk factors for OA [13]. The chondrocyte clock is not only a critical regulator of the healthy chondrocyte phenotype in mature chondrocytes [14], but has also been demonstrated to be one of the drivers of the chondrogenic differentiation pathways [9,15,16]. The chondrocyte clock is entrained by mechanical cues through daily loading/unloading patterns [13,16]. However, the upstream and downstream effectors that modulate chondrocyte homeostasis via the molecular clock are incompletely understood.

One such potential regulator is hypoxia-inducible factor-1 (HIF-1), which is a heterodimer transcription factor composed of HIF-1α and HIF-1β subunits. HIF-1α, which is activated by hypoxic conditions, is a key regulator of chondrogenesis [17]. There is now evidence that HIF-1α-mediated signaling is coupled to circadian clock synchronization in chondrocytes [18], but the specific mechanism(s) behind this crosstalk are not fully understood. Therefore, the purpose of this narrative review is to offer a synoptic analysis of the current knowledge in the field. We discuss different possible hypotheses describing the interplay between HIF-1α and the chondrocyte clock and offer some related therapeutic strategies in cartilage disorders such as OA.

For this narrative review, we searched PubMed for relevant primary research articles and recent review papers related to the molecular circadian clock and hypoxia-inducible factor-1α (HIF-1α), primarily in the field of cartilage/chondrocyte and osteoarthritis research.

## 2. The Molecular Clock in Cartilage Development

Every known organism on planet Earth is characterized by rhythmic patterns in its biological activities. Diurnal changes in mammalian behavioral patterns, metabolic actions, and physiological processes show a specific periodicity [19]. These ~24 h cycles are governed by the intrinsic molecular circadian clockwork. The endogenous clock is regulated by various Zeitgebers or time cues [20]. The presence and absence of sunlight is the main regulatory factor of the daily biological rhythms [21]. The suprachiasmatic nucleus (SCN) in the hypothalamus is the central pacemaker of the circadian mechanisms, which is light-sensitive as it receives afferent fibers directly from the retina through the retinohypothalamic tract [22]. The central timing signals generated by the SCN are transferred to every tissue in the body, leading to the synchronization of the peripheral cells’ autonomous molecular clocks [23].

The primary oscillator of the mammalian molecular clockwork is the transcriptional/translational feedback loop (TTFL) [24]. The core clock genes of the TTFL are divided into two types of interlinking mechanisms. During the positive feedback loop, the heterodimer of two transcription factors, aryl hydrocarbon receptor nuclear translocator-like ARNTL/BMAL1 (BMAL1) and circadian locomotor output cycle kaput (CLOCK), bind to the E-box sequence in the promoter region of certain downstream genes and stimulate their expression, particularly in the morning [24,25]. These factors include clock genes that are characteristic for the negative feedback loop (period (*PER1-2-3*), cryptochrome (*CRY1-2*), *REV-ERB* (nuclear receptor subfamily 1 group D member 2 [*NR1D2*])) and a number of clock-controlled genes (*CCGs*) which direct tissue-specific gene expression processes [26,27]. Later in the day, the large amount of newly translated PER and CRY proteins inhibit the function of the positive feedback loop by interacting with the BMAL1/CLOCK complex, thus suppressing their own transcriptional activity (Figure 1) [28,29].

Molecular circadian clocks exist in nearly every mammalian peripheral organ and tissue, such as the liver, pancreas, or adipose tissue [30,31]. There is evidence that hyaline cartilage is no exception, and differentiating and mature chondrocytes also express the key clock-specific transcription factors at the molecular level, both in vivo and in vitro [13,32,33]. The chondrocyte clockwork is driven by systemic cues because hyaline cartilage is a peripheral tissue and is not directly influenced by daylight. Instead, systemic signals may originate from biochemical stimuli (cyclic dietary intake that changes the metabolic activity, e.g., the availability of glucose), biomechanical stimuli (locomotion activities such as dynamic compressive mechanical load), or temperature stimuli (diurnal temperature changes in the body) [14,34]. However, the chondrogenic differentiation of certain stem cells can be influenced by the application of photobiomodulation [35], which suggests that if some of the peripheral joints (such as joints of the hand) are not completely dark, the tissues and cells of the synovial joint, including articular cartilage, may react to specific wavelengths of light.

In a human in vitro model for cartilage formation, core clock genes were found to be inactive in undifferentiated embryonic stem cells; however, upon chondrogenic induction, the key regulatory proteins of the TTFL were detected in the differentiating chondrocytes, and an oscillating expression pattern was distinguished after synchronization [36]. In micromass cultures established from chondroprogenitor cells isolated from chicken embryonic limb buds, a functional circadian clockwork was identified after applying serum shock as a clock-resetting method. Not only the clock-specific *BMAL1*, *PER2-3*, and *CRY1*-*2*, but also the cartilage-specific SRY-box transcription factor 9 (*SOX9*), aggrecan (*ACAN*), and collagen type II alpha 1 chain (*COL2A1*) genes were expressed in a rhythmic oscillatory manner. Additionally, this type of synchronization had a stimulatory effect on chondrogenesis [15]. Upon the knockout of *BMAL1* in primary chondrocytes, the expression of the hypertrophic chondrocyte-specific matrix metallopeptidase 13 (*MMP13*) and RUNX family transcription factor 2 (*RUNX2*) genes was significantly upregulated [37]. Conversely, PER1 was found to be a negative regulator of chondrogenesis. SOX6 and type II collagen protein levels were elevated after *PER1* knockdown in chondrogenic ATDC5 cell cultures [38].

Despite the accumulating data, it is still not clearly known how circadian rhythm is established during the early steps of cartilage formation.

## 3. HIF-1α: A Master Regulator of Cartilage Development

While most cells of the human body require a higher (ranging from approximately 7.5% to 4%, depending on the tissue [39]) concentration of oxygen for cellular respiration and energy production (also termed normoxia or ‘physoxia’), chondrocytes, the main cell type of cartilage tissue, reside at lower oxygen levels—this is known as hypoxic condition (O_2_ < 6%). The adaption to this special environment initiates during chondrogenesis and is primarily mediated by HIF-1 [40,41]. HIF-1 is one of the best-known transcription factors induced under hypoxic circumstances. The heterodimer HIF-1 possesses two different subunits; HIF-1α is oxygen-sensitive and is therefore stable in hypoxia, while HIF-1β (aryl hydrocarbon receptor nuclear translocator (ARNT)) is stable in normoxia [17]. Although they are constitutively expressed, in normoxia, HIF-1α is degraded via oxygen-sensitive prolylhydroxylases. During this, the prolylhydroxylase-domain-containing proteins (PHDs) hydroxylate HIF-1α, which, in this way, can be targeted by the von Hippel–Lindau protein (VHL), part of the E3 ubiquitin–ligase complex [42]. This leads to the ubiquitination of HIF-1α, making it a target of the 26S proteasome for degradation. In contrast, in hypoxia, PHDs are repressed and therefore cannot hydroxylate HIF-1α, making it stable and able to combine with HIF-1β to form the HIF-1 heterodimer. After being transported into the nucleus, HIF-1 acts as a transcription factor and binds to genes with a hypoxia response element (HRE) in their enhancer and promoter regions [43,44].

In response to hypoxia, HIF-1 triggers metabolic adaptions in the cell, including the increased expression of glucose transporters and glycolytic enzymes. Genes with various roles in apoptosis, cell proliferation, angiogenesis, and erythropoiesis are also activated by HIF-1α [45,46,47]. At the same time, HIF-1α has essential roles in numerous physiological and pathological cell processes like tumorigenesis, inflammation, and tissue development including chondrogenesis [41]. In addition to the most well-known factors that affect cartilage development, such as TGF-β, Wnt, SOX5, SOX6, and SOX9 [48,49], HIF-1α also has a pivotal role in chondrogenesis, by linking chondrocyte cell cycle with hypoxic conditions [50].

Due to the lack of vasculature, cartilage tissue develops in hypoxia. For chondrocytes to acclimatize to and survive in such a hypoxic environment, the HIF-1 transcription factor is essential. HIF-1 can influence the homeostasis of cartilage tissue by regulating the cell metabolism. Phosphoglycerate-kinase 1 (PGK) mRNA levels affect cell death in the growth plate in HIF-1α^+f/−^;colIIcre or HIF-1α^+f/+f^;colIIcre null animals. In wild-type animals, PGK is present throughout the entire growth plate and has a higher expression level in the upper hypertrophic chondrocytes, where the environment is the most hypoxic. In the null animals, the level of PGK is lower. Moreover, the knock-out animals are smaller than the controls and there is a remarkable shortening of the forelimbs and the hindlimbs. However, vascular endothelial growth factor (VEGF) expression is also dependent on regulation by HIF-1, which may contribute to the changes in the growth plate via the vascularization of the tissue [51,52].

HIF-1α is not only involved in the survival of chondrocytes in hypoxia, but also impacts molecules and pathways that influence the cell cycle, thus controlling cell proliferation and differentiation [52,53]. Moreover, HIF-1α also has an impact on extracellular matrix (ECM) synthesis (Figure 2). Among HIF-1α null and wild-type chondrocytes, the ability to produce ECM components changes according to the presence of normoxic or hypoxic conditions. Not only are the mRNA expression levels of ECM components such as aggrecan and type II collagen significantly higher, but the protein levels of these ECM elements are also considerably elevated in the wild-type chondrocytes in hypoxic condition compared to the HIF-1α null chondrocytes. In some cases, however, the same could be observed in normoxic conditions: the expression levels of *COL2A1*, for example, were statistically higher in the wild-type compared to the HIF-1α null cell cultures [54].

## 4. Interplay between HIF-1α and the Molecular Clock

In recent years, an increasing number of studies aimed to identify interactions between the circadian clock and HIF pathways [24,55,56]. Crosstalk between the circadian clock and hypoxia-regulated pathways was predicted by the CircaDB online database [57]. Hypoxia can upregulate the expression of *PER1*, *PER2*, and *CLOCK* in mouse brain and human renal cancer cell lines [58,59]. More specifically, the oxygen-sensing region of HIF-1α regulates the gene expression of *BMAL1/MOP3* and *CLOCK* in various neonatal and adult murine tissues and the human hepatocellular cell line PLC/PRF/5 [55,60,61]. The hypoxia-response element (HRE), the DNA locus to which the HIF-1α:HIF-1β heterodimers bind, is present in the promoter region of differentiated embryonic chondrocyte-expressed genes 1 (*DEC1*) and 2 (*DEC2*), which inhibit BMAL1 [62,63]. This means that hypoxia, through HIF-1α, can activate the negative regulators of the circadian rhythm (*PER1*, *PER2*, *DEC1*, and *DEC2*) in various tissues, leading to the suppression of BMAL1/CLOCK heterodimers.

Interaction between these networks is considered to be not unidirectional. The strongest induction of hypoxia target genes was observed during PER2 peak time in the liver, kidney, and heart [64]. The importance of PER2 is further substantiated by mathematical modeling, implying that PER2 plays the most crucial role in setting the period of the circadian clock [65]. The hypoxia-induced growth factor VEGF is inhibited by PER2 and CRY1, which results in periodical fluctuations in its gene expression [66]. Recently, CRY1, but not CRY2, was found to negatively regulate HIF-1α and HIF-2α via specific protein–protein interactions in mouse embryonic fibroblasts (MEF) [67]. Similar results were reported in mouse muscles: CRY1 and CRY2 suppressed HIF-1α:BMAL1 heterodimers [68]. Conversely, others have observed that PER2 and CRY1 separately enhanced HIF-1α activity by facilitating HIF-1α recruitment to the enhancer region of its downstream genes, without influencing its expression levels in HeLa cells [69]. Differences between cell types (MEFs, myotubes, and cervical cancer cells), and different CRY binding regions on HIF1 could provide an explanation for this. Apparently, the co-regulation of hypoxia and circadian pathways is more complex, due to the similarity between BMAL1 (ARNTL) and HIF-1β (ARNT) protein structures, which can each form heterodimers with HIF-1α. Low levels of BMAL1 in MEFs decreased HIF-1α accumulation in hypoxia [70]. In line with this, the co-expression of BMAL1 and HIF-1α increased the expression of HRE genes, as an adaptation to the anaerobic metabolism [71,72].

As the previous findings suggested, there is an approximately 30–50% overlap among HIF-1α and BMAL1 target genes [64]. In addition, the BMAL1/HIF-1α heterodimer also stimulates *Per2* transcription in C2C12 myoblasts, further reinforcing the connection between the circadian and hypoxia-related networks. Interestingly, in embryonic stem cells, BMAL1/HIF-1α complexes could not be observed [73]. In macrophages, the BMAL1/CLOCK heterodimer induces *NRF2* transcription factor expression, which suppresses reactive oxygen species (ROS), and therefore HIF1-α, and thus dampens the production of the proinflammatory cytokines IL-1β and IL-6 [74,75,76]. Notably, *NRF2* does not appear to be regulated by circadian clock proteins in all tissues, for instance, in the cerebral cortex [77]. Another possible bridge in the intertwined relationship between the molecular clock and hypoxia is the sirtuin protein family, which are NAD^+^-dependent deacetylases. The co-factor of SIRT3 is under circadian control, and the absence of SIRT3 in mitochondria increases ROS levels, resulting in HIF1-α stabilization in SIRT3 KO HEK293T and MEF cells [78]. When HEK293 and MEF cells were exposed to a weak-pulsed electromagnetic field, this elevated ROS production due to changes in the expression of CRY [79]. Another interesting link is via casein kinases (CK1 δ/ε, CK2). In hypoxic conditions, they can post-translationally modify BMAL1 in hamster SCN cells, and in HEK293 and MEF cells [80,81], and HIF-1α in HeLa cells [82], which may strengthen the interconnection between circadian and hypoxic regulation.

There is increasing evidence of crosstalk between HIF and circadian pathways in various in vivo animal models and clinical observations. Daily rhythms of tissue oxygenation were identified in rodents, which synchronize the molecular clocks in an HIF-1α-dependent manner, and the modulation of oxygen concentrations accelerates the recovery from jet lag in a mouse model only if HIF-1α expression is intact [83]. Circadian and HIF-1α pathways can enhance carcinogenesis and tumor progression in glioblastoma [84] and renal carcinoma [85]. Inhibition of the CLOCK–OLFML3–HIF-1α–LGMN–CD162 axis increases CD8+ T-cell-mediated immune response in glioblastoma [84]. Obstructive sleep apnea reduces blood oxygen concentration, which activates the HIF pathways, and therefore disturbs the expression of circadian clock proteins [86], which is speculated to result in metabolic and cardiovascular diseases [87]. Dysregulated activation of the NF-κB pathway may lead to chronic diseases, such as OA [88]. According to recent results, hypoxia increased NF-κB and proinflammatory cytokine activity; however, after *CLOCK* silencing, hypoxia-induced inflammatory activity was subdued [89]. It is possible that the presence of the CLOCK protein may be a requirement for the inflammatory response caused by hypoxia, which can lead to multi-systemic inflammatory disorders, such as OA. This provides evidence for an intriguing, multi-directional crosstalk between the circadian clock, hypoxia, and the immune system, also highlighting the fact that the timing of the administration of pharmaceuticals (especially for ischemic and hypoxia-associated diseases) is crucial to maximize their efficacy [90].

As described above, cartilage is time-sensitive. As a result of aging and chronic inflammation, the autonomous circadian rhythms were dysregulated in mouse cartilage [91,92,93]. In mouse OA cartilage, the downregulation of HIF-1α and upregulation of PHD2 was observed [18], which is consistent with an earlier study, where hypoxia maintained normal cartilage homeostasis [94]. The disruption of circadian clock protein expression [18] or HIF-1α depletion [95] can lead to the degradation of cartilage ECM via the activation of matrix-metalloproteinase-13 (MMP-13). Additionally, PER2 was found to respond to dimethyloxalylglycine (DMOG, a PHD inhibitor; in other words, an HIF pathway activator) in an HIF-1α-dependent manner. Furthermore, an in vivo study described a connection between circadian rhythms and the balance of collagen anabolism and catabolism [96]. The regulation of HIF-1α by BMAL1 was also confirmed in the mouse intervertebral disc; the inhibition of BMAL1 led to a reduced matrix-to-cell ratio in the nucleus pulposus, resulting in shorter discs during development [97]. In primary murine chondrocytes, the downregulation of BMAL1 dampened HIF-1α, HIF-2α, and VEGF expression, while its upregulation had the opposite effect [37]. An increased number of apoptotic cells in *BMAL1* knocked-out primary chondrocyte cultures was observed, which can be partly explained by the inhibition of HIF-1α and VEGF expression, whereas HIF-1α can regulate pro- and antiapoptotic genes [44,98].

Some of the molecular components of hypoxia-mediated signaling pathways and their interactions with the molecular circadian clock are illustrated in Figure 3.

## 5. Implications of HIF-1α Dysregulation in Cartilage Disorders

As demonstrated above, HIF-1α plays an important role in chondrogenesis and in the physiological lifecycle of articular cartilage. However, HIF-1α-related malfunctions, from excessive induction to abnormal downstream signaling, are widely reported in cartilage-related pathologies [99,100]. Here, we focus only on the two most-documented fields: OA and chondrosarcoma.

Inflammation is a key element in the pathogenesis of OA [101]. Articular chondrocytes are partially deprived of oxygen as a result of the inflamed synovial membranes using more oxygen [102]. Synovial fluid samples from OA patients demonstrated significantly lower oxygen concentrations compared to healthy samples [103]. Comparing normal and OA joints, detectable differences in the levels of HIF-1α protein expression can be observed, irrespective of the location of chondrocytes within the articular cartilage. Hypoxic conditions, catabolic stress, and IL-1β are the main factors that can increase HIF-1α accumulation in chondrocytes, delaying the rapid progression of OA at the early stages [47,104]. HIF-1α plays an essential role in directly increasing the expression of the SOX9 transcription factor [105]. Another option to elevate SOX9 levels is via BMP2. The stimulation of osteo- and chondrogenic differentiation by BMP2 was demonstrated in C3H10T1/2 mesenchymal stem cells in an HIF-1α-dependent manner [106]. In human articular chondrocytes, through the upregulation of *SOX9*, the expression of *COL2A1* and *ACAN* is increased, and chondrocyte differentiation is promoted [107]. Simultaneously with the positive regulation of *COL2A1* and *ACAN*, the inhibition of collagen types I and III also occurs in human articular chondrocytes in an HIF-1α-dependent manner [108]. Through HIF-1α, chronic hypoxic conditions significantly decrease the expressions of ADAMTSs and MMPs in chondrocytes in OA [109], specifically affecting MMP-1 and MMP13 expression levels [110]. Despite the chondroprotective effects of HIF-1α, the newly synthesized matrix components are markers of early alterations in OA cartilage. In addition to the previously mentioned examples, the delay of OA progression has other HIF-1α-related maneuvers. As observed in a transgenic mouse model, HIF-1α can maintain ATP production via the induction of VEGF expression during oxygen-limited circumstances; thus, anaerobic glycolysis allows for metabolic adaptation for chondrocytes during hypoxia [111]. The anti-catabolic responses related to *GLUT1* and *PGK1* genes are key targets of HIF-1α in early OA, as demonstrated by a comparative histological analysis of normal and OA human cartilage tissue samples [112]. Further mechanisms targeted by HIF-1α supporting chondroprotection in early OA are pathways involving anti-apoptotic responses and autophagy [44]. The inhibition of HIF-1α expression in N1511 mouse chondrocytes significantly reduced the activities of catalase and superoxide dismutase, and decreased the expression of Bcl and Bcl-xL [113]. HIF-1α may also affect the activation of caspase-8, AMPK, and mTOR in N1511 cells, suggesting that it serves a chondroprotective function by interacting with the apoptotic and autophagic responses under chronic hypoxic conditions [114,115].

Oxygen levels in tumor niches determine disease progression as cellular responses to hypoxia mainly promote neoplastic evolution [116]. In all skeletal tumors, the common cellular mechanisms affected by HIF-1α are cell proliferation, tissue vascularization and metastasis formation [117,118,119]. Although cartilage-related malignancies are referenced in a rich literature regarding hypoxia, the role of HIF-1α is also implicated in enchondromas [100]. Isocitrate dehydrogenase 1 mutations were reported to induce HIF-1α and consequently influence angiogenic properties and tumorigenicity in the JJ012 human chondrosarcoma cell line [120]. Angiogenesis is promoted by VEGF, but HIF-1α is not the only factor that stimulates VEGF expression in chondrosarcomas [121]. Patient-derived high-grade chondrosarcoma samples demonstrated increased HIF-1α expression levels associated with the upregulation of Bcl-xL. The survival rate of patients was reciprocal with HIF-1α positivity, suggesting prognostic roles of HIF-1α in chondrosarcoma [122]. The potential for HIF-1α to become a prognostic marker was further strengthened by a meta-analysis on bone tumors that revealed a significant correlation between the overexpression of HIF-1α and overall/disease-free survival [99].

In the field of molecular oncology, numerous studies reported connections between hypoxia signaling and the circadian clock. Depending on the cell type, the circadian clock exerts tumor-promoting or tumor-suppressing qualities. In glioma, loss of function mutations were identified in *CLOCK*, *CRY2*, *FBXL3*, *FBXW11*, *NR1D2*, *PER1*, *PER2*, *PER3*, *PRKAA2*, *RORA*, and *RORB* genes [123]. Loss of function mutations in these tumor-suppressive genes were negatively correlated with HIF-1α target genes (*CA9*, *VEGFA*, and *LDHA*) in glioma [123]. In colorectal carcinoma tissues, *CLOCK* showed a strong positive correlation with *HIF-1* and *VEGF* expressions [124]. CLOCK interacted with HIF-1α to enhance VEGF mRNA expression, thus augmenting angiogenesis and promoting metastasis formation [124]. Components of the molecular clock machinery were described by various studies in osteosarcoma cell lines [125,126,127,128], but not in chondrosarcoma cell lines. Although experimental evidence has been collected regarding the circadian clock in skeletal tumors, the potential crosstalk between hypoxia- and circadian-clock-related signaling remains elusive.

## 6. Therapeutic Approaches Targeting HIF-1α for Cartilage Repair

The repair of articular cartilage has been a major challenge to date due to its heavily limited regenerative capacity. Traumatic or degenerative damage to this specialized tissue represents a significant clinical burden on the health care system [129]. Recent research suggests that HIF-1α is a promising target for therapeutic interventions aspiring to repair and regenerate cartilage [130].

As described earlier, HIF-1α, a transcription factor responding to hypoxic conditions, plays a pivotal role in cellular adaptation to low oxygen levels [131]. In chondrocytes, the activation of HIF-1α has the potential to control both autophagy [114] and apoptotic processes [115,132]. Additionally, it can reduce the synthesis of inflammatory cytokines, manage the ECM environment of chondrocytes, and uphold the chondrogenic phenotype [133]. This phenotypic control extends to the regulation of glycolysis and the mitochondrial function associated with OA, ultimately leading to the formation of a more compact collagen matrix that delays the degradation of cartilage. Consequently, targeting HIF-1α presents a promising avenue for potential therapies in OA by modulating both chondrocyte inflammation and metabolism [130]. At the same time, in laboratory studies, it has been demonstrated that VEGF prompts the proliferation of chondrocytes while also triggering the expression of MMP13 through the induction of HIF-1α [134].

Various possible therapeutic strategies have emerged to modulate HIF-1α activity for cartilage repair. One such approach involves pharmacological agents, such as the hypoxia-mimetic agent cobalt chloride. In human mesenchymal stem cells (MSCs), cobalt chloride mimics hypoxic conditions in vitro by stabilizing HIF-1α. The outcomes of the experiments by Teti et al. indicated that cobalt chloride did not impact cell viability. However, the increase in chondrogenic markers such as *SOX9*, *COL2A1*, *VCAN*, and *ACAN* relied on the specific cellular origin, with most results being quite promising [135]. The prolylhydroxylase (PHD) inhibitor DMOG also stabilizes HIF-1α. Hu et al. studied the effects of the DMOG-increased expression of HIF-1α in a DMM mouse model and found that it could alleviate apoptosis and senescence via mediating mitophagy in the chondrocytes [136]. The above studies have demonstrated the these agents’ potential for cartilage regeneration.

As gene-therapy-based approaches are earning more and more attention, enhancing HIF-1α expression by techniques including viral vectors or gene editing tools has a definite appeal. Okada et al. already demonstrated that gain-of-function of HIF-1α in primary chondrocyte cultures can suppress catabolic genes such *HIF2A*, which is a direct transcriptional target of NF-κB, and induces various catabolic factors, including *MMP13*, to subsequently accelerate cartilage degeneration [137]. In this sense, HIF-2α has the opposite effect on HIF-1α. It serves as a regulator in cartilage degradation, governing the expression of numerous catabolic elements such as matrix-degrading enzymes and inflammatory mediators [138].

Platelet lysate is also gaining increased attention due to its various favorable properties for regenerative medicine. In chondrocyte cultures, platelet lysate promoted the upregulation and nuclear transport of HIF-1α, and its binding to HRE [139]. This suggests that HIF-1α could be at least partially responsible for the regenerative effects of platelet lysate. Conversely, platelet lysate may become an easily accessible and safe method for HIF-1α activation. Despite the promising effects of platelet lysate on cartilage regeneration through HIF-1 signaling, there are several limitations and drawbacks to consider. One such limitation is the variability in platelet lysate composition, which can affect its bioactivity and therapeutic efficacy. The preparation methods and donor variability can result in inconsistent levels of growth factors and cytokines in platelet lysate, impacting its ability to consistently activate HIF-1α signaling pathways [140].

Future targets may include various interacting partners that all influence the stability, and therefore the protein levels, of HIF-1α [141]. Prolylhydroxylase-domain-containing protein 2 (PHD2) utilizes oxygen and α-ketoglutarate as substrates to carry out the hydroxylation of HIF-1α. N-acetylglucosamine transferase (OGT) stabilizes HIF-1α by reducing α-ketoglutarate levels [142]. Recent findings indicate that HAUSP (USP7) acts as a deubiquitinase for HIF-1α [143]. Under hypoxia, HAUSP undergoes K63-linked polyubiquitination by HectH9, enhancing its ability to deubiquitinate HIF-1α and acting as a scaffold for HIF-1α-induced gene transcription [144]. Plasmacytoma variant translocation 1 (PVT1), a long non-coding RNA, plays an oncogenic role in various cancers. Lysine acetyltransferase 2A (KAT2A) is a histone acetyltransferase. Studies reveal that lncRNA PVT1 stabilizes HIF-1α through KAT2A [145]. Furthermore, research demonstrates that the STAT3 protein competes with pVHL, binding to HIF-1α, and consequently increasing HIF-1α protein levels [146]. GATA-binding protein 3 interacts with both full-length and the N-terminal section of HIF-1α (aa 1–401) during hypoxia, inhibiting the ubiquitination of HIF-1α [147].

Nevertheless, challenges persist in translating these findings into clinically viable therapies. Safety concerns, the precise control of HIF-1α activity to avoid undesired effects, and its long-term efficacy in clinical settings require further investigation. Additionally, optimizing activation methods and understanding the interplay between HIF-1α and other signaling pathways in chondrocytes are crucial to refine these approaches. Regardless, therapeutic strategies targeting HIF-1α hold immense promise for cartilage repair and regeneration.

## 7. Future Perspectives and Research Directions

The intricate orchestration of the molecular pathways underlying cartilage development involve a delicate interplay between HIF-1α and the molecular clock machinery [148,149]. HIF-1α stands as a central mediator in cellular responses to hypoxia, modulating an abundance of chondrogenic genes [17]. Studies have unveiled the significance of HIF-1α in promoting chondrogenesis by regulating key transcription factors, including SOX9, that are essential for cartilage-specific gene expression [44]. Furthermore, key molecules in the hedgehog pathway contain an E-box motif in their promoter regions, making them binding targets of the BMAL1/CLOCK heterodimer complex [150], or, hypothetically, even HIF-1α [151]. The hypoxic microenvironment within developing cartilage activates HIF-1α, accentuating its role in steering MSCs towards the chondrogenic lineage. As described above, accumulating evidence sheds light on the intricate relationship between HIF-1α and the molecular clock machinery. Clock genes, such as *BMAL1*, *CLOCK*, and *PER*, exhibit regulatory control over HIF-1α activity, implicating the circadian rhythm in the modulation of hypoxia-driven chondrogenesis. Even more interestingly, HIF-1α appears to reciprocally influence the expression of clock genes [86], suggesting a bidirectional interaction between HIF-1α signaling and the molecular clock [9,15,16] in the context of cartilage development.

While current research has elucidated the impact of HIF-1α and the molecular clock on cartilage development, there are numerous compelling avenues for further exploration. Deciphering the exact mechanisms governing the interaction between HIF-1α and specific clock components in chondrogenesis remains a critical area for ongoing research. Examining how external stimuli, such as changes in the environment or pathological conditions, influence this intricate molecular network offers the promise of valuable insights into disorders related to cartilage. A comprehensive understanding of the nuanced regulation of chondrogenesis by HIF-1α and the molecular clock has the potential to open new avenues for therapeutic strategies. Targeted interventions that manipulate either *HIF-1α* or clock gene expression have the potential to advance cartilage regeneration and alleviate degenerative cartilage diseases. Utilizing the capabilities of these molecular regulators may pave the way for innovative approaches to treating cartilage-related pathologies in the future.

## 8. Conclusions

The convergence of HIF-1α signaling and the molecular clock in shaping the landscape of developing cartilage unveils a captivating network of molecular interactions. In addition, due to the multi-directional crosstalk between the circadian clock, hypoxia, and the immune system, the timing of therapeutic interventions is crucial to maximize their efficacy.

Still, there are many unaddressed questions in this context. These include the broader context of signaling molecules regulated by these pathways, and the multi-directional interconnections, including the immune response and inflammatory moieties. Furthermore, there is not yet solid evidence of whether these molecular changes in circadian rhythm and hypoxia response are the causes or consequences of cartilage development and cartilage diseases such as OA. Further exploration of these intricate interplays would not only deepen our understanding of cartilage biology but also holds transformative potential in advancing therapeutic interventions for cartilage-related disorders. The stabilization of HIF-1α can alleviate hypoxia-induced apoptosis, senescence, and matrix degradation in chondrocytes. Additionally, HIF-1α influences collagen synthesis and maturation, leading to a denser collagen matrix that hinders cartilage degradation. Finally, resetting the circadian clock through interactions with HIF-1α and clock proteins presents a promising avenue for therapeutic interventions in cartilage disorders.

## Figures and Tables

**Figure 1 cells-13-00512-f001:**
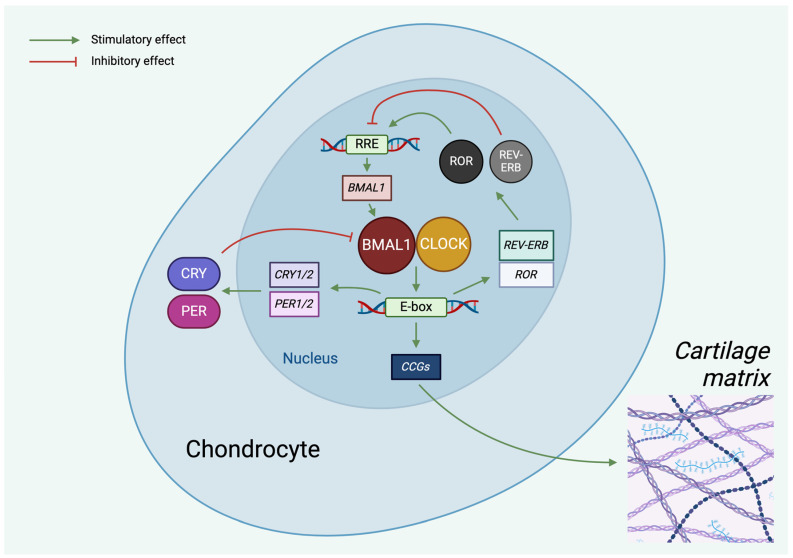
Molecular machinery of the circadian clock in chondrocytes. The circadian oscillator consists of two interconnected TTFLs. In the core loop, the CLOCK/BMAL1 heterodimer induces E-box-mediated transcription of *PER* and *CRY*, the negative regulators. PER and CRY proteins, in turn, repress E-box-mediated transcription. CLOCK and BMAL1 also control the expression of RORs and REV-ERB in the auxiliary loop, which modulate *BMAL1* mRNA levels. The rhythmic activity of the clock components also determines the expression of clock-controlled genes (CCGs). Green lines indicate stimulatory (positive), whereas red lines indicate inhibitory (negative) effects. See abbreviations in text. Created with BioRender.com.

**Figure 2 cells-13-00512-f002:**
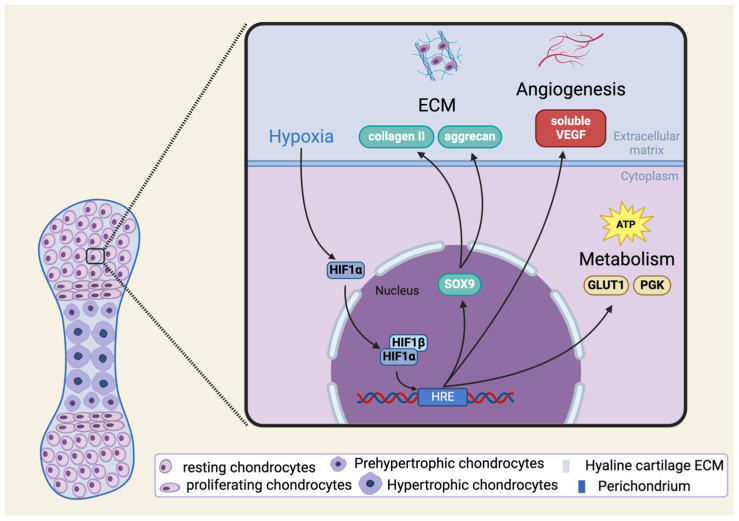
Schematic illustration of HIF-1α signaling in chondrogenesis and in mature chondrocytes. Please note that the pathways shown in the figure are not exhaustive. See abbreviations in text. Created with BioRender.com.

**Figure 3 cells-13-00512-f003:**
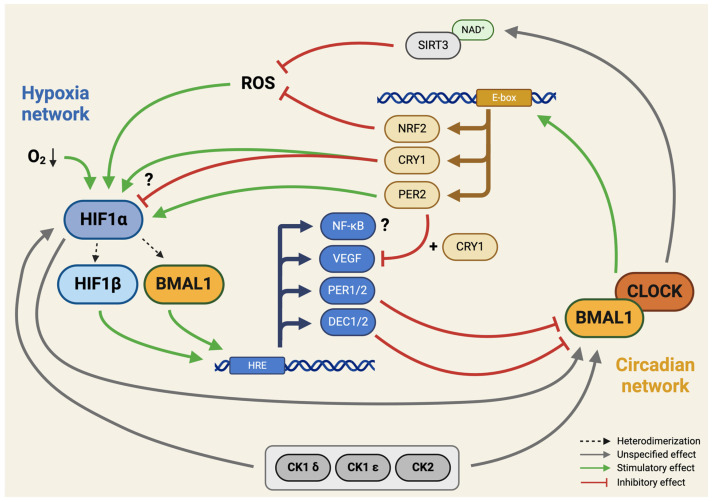
Integration of hypoxia signaling and the molecular circadian clock. This schematic illustration highlights the identified interaction sites between hypoxia signaling pathways and the molecular circadian clock in a hypothetical developing chondrocyte. The intertwining of these regulatory networks suggests a complex interplay, influencing various cellular processes and contributing to the coordination of responses to both hypoxia and circadian cues. Please note that the molecular components of pathways shown in the figure are not exhaustive. Green lines indicate stimulatory (positive), whereas red lines indicate inhibitory (negative) effects. Connections with unspecified effects are shown with grey lines. Arrowheads indicate directionality (where relevant). Question marks indicate unconfirmed (hypothetical) connections. See abbreviations in text. Created with BioRender.com.

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
