# Peer review of "Hypoxic Conditions Modulate Chondrogenesis through the Circadian Clock: The Role of Hypoxia-Inducible Factor-1α"

_cells, 2024, doi:10.3390/cells13060512_

Round 1
Reviewer 1 Report
Comments and Suggestions for Authors
This review entitled "Hypoxic Conditions Modulate Chondrogenesis Through the Circadian Clock: The Role of HIF-1α" aimed to clarify the role of HIF-1α in the cartilage formation and physiological life cycle of articular cartilage which is in the hypoxic condition. Authors used published evidences to interpret the interaction between the HIF pathway and the circadian clock. They report HIF-1 is a major regulator of cellular responses to hypoxia by activating transcription of genes that promote metabolic adaptation to hypoxia. In addition, authors provide a concise analysis of current understanding of the dynamic interplay between HIF-1α and the chondrocyte molecular clock in health and disease, especially, they incorporate innovative interpretations and propose related treatment strategies for osteoarthritis and other cartilage diseases with HIF-1α as the main axis. Moreover, authors deduce different hypotheses on the intricate interactions with HIF-1α regulatory pathways, and provide new concepts and ideas in this field.
Overall, this review is a valuable reference for academic experience and clinical treatment strategy made. However, there are many programmer errors in the text that must be corrected. Additionally, the meaning of dash arrow (↓↑) in figure 1 must be showed.
Reviewer 2 Report
Comments and Suggestions for Authors
Dear Author,
The manuscript titled " Hypoxic Conditions Modulate Chondrogenesis Through the Circadian Clock: The Role of HIF-1α " explores the impact of HIF-1 α in the modulated circadian clock on chondrogenesis under hypoxic conditions. The study provides important insights into the potential role of HIF-1α that regulates the influence of circadian rhythm on the cartilage process under hypoxic conditions Overall, the manuscript is well-structured and presents a valuable contribution to the field. However, there are a few areas that could benefit from further clarification and expansion.
Specific Comments:
Title:
The title is good and doesn’t need any revision.
Abstract:
Line 10: "HIF-1" should be "HIF-1α" for consistency with the rest of the abstract.
Line 12: "chondrogen- esis" seems to have a hyphenated error. It should be "chondrogenesis."
Line 13: "interaction" should be "interactions."
Line 15: "suggest" should be "suggests."
Line 20: "cre-" seems to have a hyphenated error. It should be "creative."
Line 22: "osteoar- thritis" seems to have a hyphenated error. It should be "osteoarthritis."
Line 23: Semicolon at the end of the abstract should be a period for clarity.
Line 15-18: ‘’ Although the circadian clock is an emerging regulator in both developing and mature chondrocytes, how circadian rhythm is established during the early steps of cartilage formation and through what signaling pathways it promotes the healthy chondrocyte phenotype is still not entirely known”. please write in simple and concise way instead of compound sentence as it would be more appropriate for understanding.
Line 19-20: ‘’ the molecular clock in chondrocytes, both in states of health and disease, while also incorporating creative interpretations’’. Please emphasized it for better understanding.
Introduction:
Clearly state the study's objectives to provide a concise overview.
Line 28: "disease" should be "diseases" for grammatical correctness.
Line 29-30: ‘’ The tissue that is involved in transmitting mechanical load and providing smooth articulation of bones but has a limited capacity for regeneration’’ Please rephrase it for better understanding’’.
Line 30: "articulation of bones" might read better as "bone articulation."
Line 32-33 ‘’OA can affect any joint but most commonly impacts the knee, the hip, and the joints of the hand’’. Please mention what are the possible reasons that affect knee, hip joints and also discuss why OA doesn’t affect other joints of the body.
Line 34: "molecular endotypes" could be "molecular subtypes" for clarity.
Line 41: "carti- lage" seems to have a hyphenated error. It should be "cartilage."
Line 44: "why articular cartilage regeneration fails" could be rephrased as "why the regeneration of articular cartilage fails."
Line 52: ‘’Chondrogenic pathways are regulated by multiple external and internal factors’’ Please elaborate the how external and internal pathways regulate chondrogenesis.
Line 61-62:’’ Activated by hy poxic conditions, HIF-1α is a key regulator of chondrogenesis’’. Please rephrase it for a better understanding’’.
Line 77: "cell-autono- mous" seems to have a hyphenated error. It should be "cell-autonomous."
Line 89: "freshly translated PER and CRY proteins" might read better as "newly translated PER and CRY proteins" for clarity.
Line 115: "circadian rhythm" could be "circadian rhythms" for plural consistency.
Line 98-99:’’ Systemic signal may originate from biochemical, biomechanical or temperature stimuli’’. Please elaborate it for a better understanding.
Line 147-148:’’ For chondrocytes to acclimatize to and survive in such a hypoxic environment, the HIF-1 transcription factor is essential’’. Please discuss the mechanism in related molecular circadian rhythm and how HIF-1 affects the chondrocyte under hypoxic conditions.
Line 124: "transcription factors" should be "transcription factor" to match the singular "factor" in the sentence.
Line 129: "hydroxylate" should be "hydroxylates" for subject-verb agreement.
Line 134: "to" should be "of" for clarity in "HIF-1α combines with HIF-1β to form the HIF-1 heterodimer."
Line 145: "with hypoxic conditions" might read better as "in hypoxic conditions" for clarity.
Line 199: "approximately 30–50%" should have an en dash (–) instead of a hyphen (-) for range indication.
Interplay between HIF-1α and the Molecular Clock:
Line 169: "between the circadian clock and the HIF pathways" could be "between the circadian clock and HIF pathways" for conciseness.
Line 172: "moiety" could be replaced with "region" for clarity.
Line 179: "not to be unidirectional" might read better as "not unidirectional."
Line 200: "BMAL1:HIF-1α heterodimer" could be "BMAL1:HIF-1α complex" for clarity.
Line 215: "different in vivo animal models" could be "in various animal models" for clarity.
Line 215: "multi-directional" could be "bidirectional" for clarity.
Line 224: "problems" could be "issues" for variety in language.
Line 229: "interesting, multi-directional crosstalk" might read better as "intriguing, bidirectional crosstalk."
Line 231: "hypoxia-related diseases" could be "hypoxia-associated diseases" for clarity.
Line 254: "the two most-documented fields only" might read better as "only the two most-documented fields" for clarity.
Line 260: "protein expression levels" could be "levels of protein expression" for clarity.
Line 267: "man- ner" seems to have a hyphenated error. It should be "manner."
Line 278: "chondroprotection in early OA" could be "early OA chondroprotection" for clarity.
Line 289: "higher-grade" could be "high-grade" for clarity.
Line 293: "correlation between HIF-1α overexpression and overall/disease-free survival" might read better as "correlation between overexpression of HIF-1α and overall/disease-free survival" for clarity.
Line 303-304. Consider quoting
Xian et al. 2023. Chondrocyte apoptosis as a potential mechanism in ostrich limb and toe disorders: a pathological investigation. Pak Vet J. http://dx.doi.org/10.29261/pakvetj/2023.119”
as it highlights the role of chondrocyte apoptosis in limb and toe disorders. The findings of this investigation align with your discussion on the importance of understanding apoptotic processes in cartilage repair.
Also, incorporate this study
Zhu et al. 2023. Ameliorative effects of triptolide against autophagy and apoptosis in thiram induced tibial dyschondroplasia. Pak Vet J, 43(1): 132-138. http://dx.doi.org/10.29261/pakvetj/2023.008
Khan et al. 2022. Supplemental selenium nanoparticles-loaded to chitosan improves meat quality, pectoral muscle histology, tibia bone morphometry and tissue mineral retention in broilers. Pak Vet J, 42(2): 236-240. http://dx.doi.org/10.29261/pakvetj/2022.007
after Line 304 where the discussion on autophagy and apoptosis is already present. You can mention how the findings of this research support the potential therapeutic benefits of targeting these pathways for cartilage repair Line 304 where the discussion on autophagy and apoptosis is already present
Line 335-336: ‘’In chondrocyte cultures, platelet lysate promoted the elevation and nuclear transport of HIF-1α, and its binding to HRE’’ discuss pathway signaling how platelet lysate modulate influence on cartilage regeneration by HIF-1 and discuss their limitation and drawbacks.
Line 308: "degradation of cartilage." - Add a period at the end to complete the sentence.
Line 312: "MMP- 312 13" - There's an extra space between "MMP-" and "13". Remove the extra space.
Line 320: "The prolylhydroxylase (PHD) inhibitor DMOG also stabilizes HIF- 320 1α." - There seems to be a formatting issue with an extra space and a tab. Remove the extra tab to align the text properly.
Line 376: "Please note that the list of path-" - "path-" seems to be cut off. Ensure the word "pathways" is complete.
Line 381: "See abbreviations in text." - Add a period at the end to complete the sentence.
Line 382: "While current research has elucidated the impact of HIF-1α and the molecular clock 382 on cartilage development, there are numerous compelling avenues for further explora-" - Similar to Line 376, "explora-" seems to be cut off. Ensure the word "exploration" is complete.
Line 402: "multi-directional intercon-" - "intercon-" seems to be cut off. Ensure the word "interconnections" is complete.
Line 403: "nections including immune response and inflammatory moieties, and finally to get solid" - There seems to be a lack of coherence in this part of the sentence. Consider rephrasing for clarity and coherence.
Line 407: "biology but also holds transformative potential in advancing therapeutic interventions for" - Consider specifying what type of transformative potential is meant here for clarity.
Discuss on the potential implications and strengthening of the HIF-1 impact regulated influence on chondrogenesis by physiologic condition.
Decision:
1. Overall manuscript is well structured and focuses on the current study of the field. However, a few areas could be clarified and improved for a condensed version of the review of this study.
Comments on the Quality of English LanguageThey are included in the report.
Reviewer 3 Report
Comments and Suggestions for Authors
The review manuscript covers a novel and exciting topic.
Some paragraphs are very long and could be subdivided for better readability.
The sentence in the abstract lines 15-18 is too long and should be subdivided.
Introduction: the switch from OA to cartilage tissue engineering (TE) is abrupt and it could be better explained why TE makes sense to inhibit the onset of OA.
line 75: "suprachiasmatic nucleus ... is light sensitive peacemaker" it might be misleading since the nucleus contains no light-sensitive receptors but receives only afferents from the retina...
line 94: that hyaline and articular cartilage (sounds a bit misleading since nearly all joint cartilages are hyaline except e.g. for the temporomandibular joint with some fibrocartilage)
"not directly sensitive to light" The mode of action might be a miracle but cells undergoing to chondrogenesis such as ASCs could be sensitive to light PMID: 34217028 (Schneider et al., 2021). The joint cavity might not be completely dark.
line 103: "chondrifying" is an unusual word / (micromass cultures undergoing chondrogenesis?)
please check whether the sentence in lines 129-132 is complete
line 165: ",but.." "also" could be added. Not clear to me: does the sentence mean "compared to HIF-1alph null mutant"?
line 177: "Hypoxia through HIF-1alpha can activate negative regulators of the circadian rhythm" references should be added, in which cell type was it observed and what is the physiological sense of it?
line 221: "affect" better to explain exactly in which manner
line 224: "alters" please explain in which manner
Line 226: it remains unclear how clock influences OA
line 244: Style: "damaged" compromised/inhibited?
5.
I would introduce subheadings e.g. OA and chondrosarcoma
line 280: there is a break
lines 281-295
the relation to HIFalpha but not to the circadian clock is described
cobalt: risk of allergy? line 316
Reviewer 4 Report
Comments and Suggestions for Authors
In general, the review is interesting but I have several comments.
The abstract focuses on the role of HIF-1 in chondrocyte hypoxia and the connection with the circadian clock. Only in the last part is a mention made of the fact that both healthy and pathological conditions will be discussed, and in particular OA. Moving from reading the abstract to the introduction, the reader expects an introduction on cartilage/chondrocytes and HIF, and is therefore a little confused as the authors start immediately from OA.
Lines 27-28: references are lacking.
Lines 31-32: references about cartilage/chondrocytes biomechanics in OA should be added.
Introduction about OA should be improved. It should be added that OA is a whole joint disease, involving all joint tissues such as synovial membrane, subchondral bone, meniscus and infrapatellar fat pad in addition to cartilage.
Line 33-34: references on the different OA joint that are affected are missing.
A brief section about the methods (keywords and databases) used for this review should be added after the introduction, even if it is a narrative review.
Section 2: a figure summarizing how the molecular clock works in general and in cartilage should be added.
Section 3: a figure illustrating the role of HIF-1 in hypoxia and in cartilage development should be added.
Lines 166-167: this point should be better explained.
Lines 170-172: in which tissues?
Section 4: a figure or a table summarizing the interplay between HIF-1 alpha and the molecular clock could be useful for the reader. Figure 1 seems to be more appropriate in this section.
Lines 209-211: in which cells or model?
Lines 213-215: in which cells or model?
Lines 220-222: “Circadian and HIF-1α pathways affect carcinogenesis and tumor progression by immune suppression.” Which cancer? Only glioblastoma?
Lines 239-241: reference is missing.
Line 244: human primary chondrocytes?
Lines 252-254: references should be provided.
Lines 254-255: it is unclear, why the authors reported this sentence here. Chondrosarcoma was not even mention. Chondrosarcoma is mentioned after (lines 287-295). Limitations should be moved and improved in section 7.
Line 256: references should be proved.
Lines 266-280: could the authors specify if this information is related to animal models or mouse chondrocytes or human chondrocytes etc?
Lines 281-284: references are missing.
Section 5: the link between HIF-1alpha, OA and apoptosis /autophagy should be better explained.
Lines 303-304: references are lacking.
Lines 314-316: in OA?
Lines 316-320: in human chondrocytes?
Line 328: what is the function/role of HIF2A?
Other comments:
When the authors refer to genes, gene names should be written in italics.
Round 2
Reviewer 2 Report
Comments and Suggestions for Authors
no
Comments on the Quality of English Languageno
Author Response
Thank you for approving the revised version of our manuscript.
Reviewer 3 Report
Comments and Suggestions for Authors
The author team fully addressed my concerns and modified the manuscript accordingly.
Author Response

(The authors gave the same response as above.)

Reviewer 4 Report
Comments and Suggestions for Authors
No additional comments
Author Response

(The authors gave the same response as above.)
